# Optofluidic Particle Manipulation: Optical Trapping in a Thin-Membrane Microchannel

**DOI:** 10.3390/bios12090690

**Published:** 2022-08-27

**Authors:** Zachary J. Walker, Tanner Wells, Ethan Belliston, Seth B. Walker, Carson Zeller, Mohammad Julker Neyen Sampad, S. M. Saiduzzaman, Holger Schmidt, Aaron R. Hawkins

**Affiliations:** 1Department of Electrical and Computer Engineering, Brigham Young University, Provo, UT 84602, USA; 2School of Engineering, University of California, Santa Cruz, CA 95064, USA

**Keywords:** lab-on-a-chip, biosensor, nanopore, optofluidic, microfluidic, gradient force, radiation pressure, optical trap

## Abstract

We demonstrate an optofluidic device which utilizes the optical scattering and gradient forces for particle trapping in microchannels featuring 300 nm thick membranes. On-chip waveguides are used to direct light into microfluidic trapping channels. Radiation pressure is used to push particles into a protrusion cavity, isolating the particles from liquid flow. Two different designs are presented: the first exclusively uses the optical scattering force for particle manipulation, and the second uses both scattering and gradient forces. Trapping performance is modeled for both cases. The first design, referred to as the orthogonal force design, is shown to have a 80% capture efficiency under typical operating conditions. The second design, referred to as the gradient force design, is shown to have 98% efficiency under the same conditions.

## 1. Introduction

Microfluidic bioparticle trapping has been demonstrated in many instances with electrical, magnetic, optical, and hydrodynamic being most common trapping mechanisms [1]. Electrical trapping is often done by isoelectric focusing (also known as electrophoretic focusing) which uses electrodes to manipulate molecules based on their pH levels [2]. Molecules outside their isoelectric point will have a positive or negative charge which allows them to be attracted to and trapped by an electrode [3]. Another form of electrical trapping is known as dielectrophoresis. This method generates motion of dielectric particles such as polymer beads in a non-uniform electric field. The particle will be attracted to either the positive or negative electrode based on the particle’s permittivity and will be held by the electrode until further interrogation [4]. Magnetic trapping works similarly to electrical but is implemented instead with a magnetic field. A field gradient is generated by magnets on chip and is then used to hold molecules that are bound to magnetic nanobeads [5]. Optical trapping can be implemented using on-chip counter propagating dual beam divergence and loss-based traps to hold particles for further manipulation [6,7,8]. The anti-Brownian electrokinetic (ABEL) trap is a more efficient optical trap that uses counter propagating beams in combination with electroosmotic flow to assist with centering based on fluorescent feedback [9]. Another optical trapping method is optical tweezers. A focused laser uses scattering and gradient forces acting in unison to confine the particle in the beam waist or focal point [10]. This method is advantageous as it is easy to manipulate and move the particle by translation of the laser [11]. Hydrodynamic trapping is a common method as it does not require external forces. Physical mechanisms such as walls, columns, pores, arrays or side channels have been implemented to create trapping sites based on mechanical or suction capabilities in order to trap and hold objects [12]. These trapping mechanisms can be used in biosensing applications to aid in the detection process. Biomarkers such as DNA, RNA, proteins, and viruses are bound to carrier microbeads that are easier to trap and manipulate than individual biomarkers.

This paper introduces a chip-based particle trap which utilizes both physical and optical trapping. Physical confinement is realized by injecting particle-containing fluids into microfluidic channels. These channels contain protrusion cavities where particles are meant to accumulate, isolating them from fluid flow forces. Particles are pushed into these cavities using optical forces from light introduced into the channels. This mechanism serves as a particle sorter and concentrator and can drastically increase particle concentration in the small protrusion cavities by collecting particles from a fluid flow stream. Once isolated in the cavity, these particles can be analyzed through a variety of mechanisms [13,14] or altered through chemical reactions, temperature, or mechanical stimulation [15,16].

Two device geometries are demonstrated in this paper and shown in Figure 1. The first is referred to as the orthogonal force design and uses only the optical scattering force as the particle trapping mechanism. The second design, referred to as the gradient force design, uses both the optical scattering force and the optical gradient force for particle trapping. The scattering force manipulates particles by way of radiation pressure or induced momentum from electromagnetic radiation. It has already shown to be useful in optofluidic applications for manipulating single molecules or nanobeads suspended in liquid microchannels [17,18,19]. Our “orthogonal force design” uses a liquid-core (LC) microfluidic channel for fluid and particle transport. When particles enter the optofluidic region, the photonic momentum coming from a solid-core (SC) ridge waveguide pushes on them, forcing them into a protrusion cavity and holding them against a channel sidewall as demonstrated in Figure 1a. One application of the orthogonal force design would be a system containing multiple protrusion cavities meant as a particle sorter. Different optical powers could be used to push particles of different sizes into individual cavities.

The “gradient force design” has a longer optofluidic channel as compared to the orthogonal force design and utilizes the optical scattering force and the gradient force for particle sorting. As shown in Figure 1b, the gradient force focuses particles to the center of the liquid microchannel where the optical power is the strongest [20]. The scattering forces then push particles into the protrusion cavity. Without the use of the gradient force, particles traveling close to the right sidewall would be more susceptible to the pull of the fluid stream at the channel bend preceding the protrusion cavity. Utilizing both scattering and gradient forces should make this design more efficient at particle capture for a given optical power, when compared to the orthogonal force design.

An exciting feature of this optofluidic device is that it hosts a suspended 300 nm thin membrane over the optofluidic section of the liquid channel. This membrane acts as a very thin barrier between the channel surface and fluid. This becomes advantageous for applications that combine this platform with particle analysis. One such application is the use of integrated solid-state nanopores as a means of particle identification when particles translocate the pore [21]. Thin membranes are required for focused ion beam (FIB) nanopore drilling in order to achieve tight pore dimensions [16]. The concentration and isolation functionality of this platform offers an excellent way to position high numbers of samples directly under the pore. Indeed, this was one of the major motivators for the development of the optofluidic platform and future studies will show particle trapping and nanopore analysis acting in concert.

## 2. Experimental and Methods

The fabrication process for both the orthogonal and gradient designs closely follow a previously demonstrated method [22,23]. An open channel is etched into silicon and silicon dioxide is grown on the surface through thermal oxidation. The interface between the ridge waveguide and the liquid channel is critical as the entire 2 μm silicon wall at this interface must be converted to silicon dioxide in order for light to be transmitted into the liquid channel. Next, plasma enhanced chemical vapor deposition (PECVD) is used to grow high index silicon dioxide for waveguide use. SU8-2000.5, a non-viscous photopolymer, is then introduced from both sides of the channel and the polymer flows down the channels by way of the spontaneous capillary effect. The surface tension of the silicon dioxide causes the polymer to be pinned at the top of the channel without spilling out. The result is a meniscus shaped sacrificial material that acts as a template for nanomembrane formation. The nanomembrane is then grown over the top of the SU8-2000.5 using PECVD silicon dioxide. Finally, the sacrificial material is etched and removed in a piranha (sulfuric acid and hydrogen peroxide) mixture, leaving a hollow channel with a suspended, 300 nm thick membrane, as shown in Figure 2a. The channel cross sectional dimension is 10 μm by 10 μm.

The ridge waveguide shown in Figure 2b is fabricated simultaneously with the liquid channel and has a cross sectional dimension of 10 μm by 3 μm. A pedestal is etched into silicon which forms the basis for the waveguide. Thermal silicon dioxide is subsequently grown over the pedestal followed by high index PECVD silicon dioxide. The ridge waveguide pattern is then etched into the PECVD silicon dioxide using reactive ion etching. A cladding layer of low index PECVD silicon dioxide is then added to complete the ridge waveguide. The interface between the ridge waveguide and liquid channel is critical as the original silicon must be converted to thermal silicon dioxide for optical transmission as shown in Figure 2c. A section of the finalized optofluidic region for both designs is shown in Figure 3.

## 3. Results

The optical characteristics of each design are crucial to the trapping capabilities of the platform. Optical mode simulations were performed in Lumerical FDE for both the liquid channel and solid core ridge waveguide using 532 nm as the wavelength. The zero order E field intensity mode profile distributions for both the liquid channel and ridge waveguide are shown in Figure 4a,b. The simulations are approximate and, for simplicity, do not take into account the exact meniscus shape. The gradient force is possible due to the optical power distribution in the liquid channel. This distribution creates the electric field gradient which enables particle focusing in the center of the channel where the optical intensity is the highest. In order for the optofluidic focusing section of the gradient force design to be effective, it needs to be long enough to give the gradient force enough time to pull particles to the center of the channel under typical fluid velocities.

Managing the optical loss in the various sections of the optofluidic platform is a critical parameter for high trapping efficiency. These devices do not utilize reflecting layers like those found in anti-resonant reflective optical waveguide (ARROW) structures, so the optical loss in the liquid channel is expected to be high since the aqueous solutions have a lower refractive index than the surrounding silicon dioxide walls. The average liquid channel optical loss using top view scattering was measured at 6.61 cm^−1^ while the ridge waveguide was 0.50 cm^−1^ [22]. Due to the high optical loss, this platform can only be used in proximity applications where optofluidic sections are kept short. The optical transmission directly correlates to the optical scattering force available to influence particles. The transmission for the orthogonal force design from chip edge to the protrusion cavity is calculated to be 19%. The transmission from chip edge to protrusion cavity for the gradient force design is 17.9%. Transmission values were calculated based on measured values for waveguide loss and coupling coefficients between the fiber and chip as well as between the ridge waveguide and liquid channel [22,24,25].

### 3.1. Particle Manipulation in the Orthogonal Force Design

The orthogonal force design introduces radiation pressure from the ridge waveguide orthogonal to the liquid channel. This design brings the particles as close to the ridge waveguide as possible. It can be used in applications that require an array of multiple ridge waveguides and trapping cavities where the trapping efficiency of an individual cavity is not the main priority. The orthogonal force design also allows for selectively sorting particles based on their size or refractive index. The forces acting on a particle in the optofluidic region of the orthogonal design is provided as an illustration in Figure 5. The fluid simulations were done in Ansys Fluent and are in the laminar flow regime.

The optical scattering force is the trapping mechanism for the orthogonal force design. It can be determined using
(1)$$F_{Scatter}=Q\pi \frac{na^{2}}{c}I$$
where *Q* is a dimensionless variable relating the momentum transfer efficiency, *a* is the radius of the particle, *n* is the index of refraction of the liquid, *c* is the speed of light, and *I* is the intensity of the optical power incident on the particle. This equation can be used to calculate the minimum optical power required to capture particles for various sizes and flow velocities as show in Figure 6. For the minimum power required to trap a particle, we consider the worst-case scenario where a particle is on the side wall furthest from the protrusion cavity. A particle against the wall will be moving at its slowest rate as the velocity is not constant along the channel. Our calculation uses an average fluid velocity in the x direction to account for different possible flow rates as the particle transitions towards the cavity. If the velocity induced on the particle by the scattering force can match the velocity of the fluid flow, the particle will be trapped. The velocity produced by the scattering force is found by considering the matching drag force on the particle (which is velocity dependent). Because of the square geometry formed by the 10 μm waveguide and 10 μm channel width under consideration near the protrusion cavity entrance; a particle would travel the same distance in the y direction as in the x to be caught. We are assuming a terminal velocity due to the short span of the acceleration. These calculations are based on an approximation where the power is equally distributed vertically. The drag force equation holds due to laminar flow inside the microfluidic channel.

Minimum trapping power was calculated for a range of particle diameters and flow velocities as shown in Figure 6. High velocity particles and smaller particle diameters require more optical power for trapping. If the minimum radiation pressure is not induced on the particle, it will pass through the optofluidic region without being contained in the protrusion cavity.

Experiments were performed using 400 mW of optical power from a laser source, 72 mW at the optofluidic trapping site due to the 19% transmission efficiency shown previously, 532 nm wavelength, and 1 μm diameter particles. The trapping efficiency can be found in the Results section. The intersecting lines and dot in Figure 6b demonstrates the power, particle size, and flow velocity expected by our experimental values. A video showing particles being trapped for these conditions can be found in Appendix A. The fluid flow for these videos was approximately 100 μm/s. Given that we are operating very close to the minimum optical power necessary to trap particles, we do not expect all of them to trap as they flow past the protrusion cavity. The fact that the ridge waveguide does not line up vertically with the center of the liquid microchannel, thus causing uneven illumination, also leads us to expect incomplete trapping.

### 3.2. Particle Manipulation in the Gradient Force Design

The gradient force design was developed as an efficient optical trapping system. The beam propagation and liquid flow are both in the same direction and act in concert to push particles toward the protrusion cavity. The gradient force causes the particles to be centered in the channel and helps prevent particles from escaping.

The gradient force can be analyzed using Equation (2) below
(2)$$F_{grad}=\frac{2\pi n_{2}a^{3}}{c}\left(\frac{m^{2}-1}{m^{2}+2}\right)\nabla \;I$$
where $n_{2}$ is the index of refraction of the particle, a is the radius of the particle, *c* is the speed of light, m is the ratio of the index of refraction of the particle to the index of the medium and I is the optical intensity [26]. Equations (1) and (2) hold for homogeneous spheres corresponding to the microbeads used in our experimentation. The gradient force is plotted for various particle diameters versus the channel position in Figure 7, utilizing the electric field intensity as shown in Figure 4. The gradient force increases from the wall of the channel towards the center. Particles are attracted to the region where the optical intensity is the strongest. This enables particles to be pulled from anywhere in the channel to the center. As shown in Equation (2), the gradient force is directly proportional to the gradient of optical intensity. Higher intensity allows for faster and stronger gradient force channel centering.

Understanding the properties of the gradient force is important in determining the optical trapping characteristics of the optofluidic manipulation chip. The effects of the gradient force on the motion of a particle can be described by the differential equation
(3)$$m\;\frac{dx^{2}}{dt^{2}}=F_{grad}$$
where m  is the mass of the particle and $\frac{dx^{2}}{dt^{2}}$ is the second derivative of the position in the x direction in the channel in terms of time (directions shown in Figure 8a). Solving this differential equation gives insight into the trapping characteristics of this device. The first condition considered is a worst-case scenario in which a particle is at the entrance of the optofluidic region on a side wall. For a fluid moving at a given speed, it is desirable to know how far up the channel in the z direction the particle would travel before becoming susceptible to the gradient force and centralizing as shown in Figure 8a. MATLAB was used to solve the differential equation numerically using the ode45 solver function with initial conditions for particle position and velocity. Figure 8b demonstrates the optical power required for centralization with corresponding travel distances based on solutions to Equation (3) for a series of spherical particle sizes (assumed to be spheres) with an index of refraction of 1.6 according to Thermo fisher. Fluid velocity was assumed to be 100 mm/s and water was used as the solution (*n* = 1.33). As shown in the figure, higher optical powers are required to center smaller particles. It is important that the particles move to the center of the channel before reaching the channel bend before the protrusion to ensure that the particles are captured in the cavity. The intersecting lines and dot in Figure 8b demonstrate the power and corresponding distance expected given the experimental parameters discussed in the Results section. Video observations confirm our centering distance predictions in Figure 8b.

The next important consideration is what happens to a particle which has been centered in the optofluidic channel and then encounters the bend near the end of the channel, where it becomes susceptible to the fluid flow moving orthogonally to the trapping direction. This is the critical junction where trapping in the protrusion cavity will either take place or the particle will be swept away by the fluid flow. Knowing the forces acting on the particle and velocity components at this point is important to ensure that enough optical force is present to capture the particle. These forces are illustrated in Figure 9. The gradient and scattering force components are the trapping mechanisms in this case.

If the particle does not overcome the channel flow and enter the protrusion cavity, it will not become trapped. Figure 10a demonstrates trapped and un-trapped trajectories. The trapped trajectory is at the minimum power threshold showing that at the bend, the particle will be pulled in the direction of fluid flow, causing the curve in the particles path. In this scenario, the particle will barely enter the protrusion cavity where it will no longer be susceptible to the fluid flow and become centered through the gradient force as demonstrated in Appendix A. The same methods explained previously were used to solve Equation (3) for this scenario. The solutions were used to determine the minimum power required to ensure trapping for given particle diameters and for various flow velocities as shown in Figure 10b. Faster flow rates and smaller particle diameters require higher power for particle trapping.

Experiments were performed by coupling 400 mW of optical power onto the chip with ~72 mW at the optofluidic trapping site due to the 17.9% transmission calculation shown previously, 532 nm wavelength, and 1 mm diameter particles and the findings can be found in the Results section. The intersecting lines and dot illustrate where the experimental parameters would appear on the graph. A video showing particles being trapped for these conditions can be found in Appendix A. The fluid flow for these videos was approximately 100 mm/s. Given that our experimental conditions are safely above the calculated optical power necessary to trap particles, we expect nearly all of them to trap as they flow past the protrusion cavity.

Solving the differential equation both at the start of the channel and at the corner proves the necessity of the gradient force as it first pushes particles to the center of the channel and then to pull them back near the turn. Solutions to Equation (3) shown in the simulation results in Figure 8b and Figure 10b makes it evident that without the minimum power from the gradient force, the particles will not become trapped. The scattering force alone will not be enough to trap particles, especially if they were near the right wall close to the corner.

The trapping efficiency for both designs presented here was experimentally tested by edge coupling light into the chip. 400 mW of optical power from a 532 nm laser source was coupled through an optical fiber and into the ridge waveguides on chip. Polystyrene microbeads with a diameter of 1 μm were used as the particles in the experiment. Metallic cylindrical reservoirs were attached to the microfluidic channels at the both the inlet and the outlet. A fluid solution containing microbeads with a concentration of 3.8 × 10^7^ beads/mL was introduced into the inlet reservoir and the fluid was gravity fed into the system. This created an average fluid flow velocity of approximately 100 μm/s. The experiment counted the total number of beads collected at the protrusion cavity versus those that escaped. Examples of trapping events are shown in Appendix A. Trapped and escaped particles were counted versus time and the results from two separately tested orthogonal devices and three separately tested gradient devices are superimposed and shown in Figure 11. Similar trends held for other tests made on twelve separate orthogonal chips resulting in efficiencies within 8% and three separate gradient chips resulting in efficiencies within 3% compared to those shown in Figure 11.

Because we are demonstrating a flow-through device capable of discriminating particles in the fluid stream, Figure 11 demonstrates what happens as particles move through over time. The performance does not significantly change during the course of the experiment due to accumulation of particles in the trapping region.

The number of particles captured divided by the total number of particles entering the optofluidic system was used to calculate the trapping efficiency. For the orthogonal design, the measured optical trapping efficiency was 80% and the gradient force design had a trapping efficiency of 98%. These results show that our models match closely with the experimental data. The orthogonal design experimental point did not have much margin for error as shown in Figure 6b, causing some of the particles to pass through the optofluidic region without capture. The margin of error for the experimental point is much higher in the gradient force design, shown in Figure 10b. This results in a more reliable trapping efficiency.

## 4. Discussion

The results from the experimental data collected demonstrate high trapping efficiency utilizing these meniscus membrane optofluidic devices. The orthogonal force design has a lower trapping efficiency but can still be advantageous in certain applications. Particle sorting based on size is possible using the orthogonal force design. Multiple waveguides and protrusion cavities can also be incorporated into this design for an array of trapping sites.

The gradient force design allows for exciting possibilities due to the high trapping efficiency. The scattering force in connection with the gradient force creates a trapping mechanism that is very reliable and consistent. The ability to centralize particles through the gradient force is critical to this particle trapping solution.

It is noted that our computations for both designs are based on simple models of what is a very complicated system. The analysis attempts to provide guidance for operation parameters involving fluid flow and laser power for high efficiency trapping. Our model gives a first order look at approximate powers where particle trapping is expected for meniscus membrane optofluidic devices. The tests performed at the maximum laser power available closely matches the behavior predicted by our simple models. While not explicitly reported, turning down the laser power results in less particle trapping.

## 5. Conclusions

The optofluidic particle manipulation chip demonstrated in this paper provides new opportunities for optofluidics, particle analysis, and biosensing. The high trapping efficiency of the gradient force design coupled with the thin nanomembrane offers possibilities for applications such as solid-state nanopores. The gradient force allows for a high optical trapping efficiency of 98% while the orthogonal force design demonstrated a trapping efficiency of 80%. Solutions to differential equations predicting particle trajectories provide important insights into the characteristics of the scattering and gradient forces acting on a given particle and can determine the optical power required for high trapping efficiency. The power and distance required to centralize a particle and the gradient force needed for capture are two important results given by solutions to the gradient force differential equation.

## Figures and Tables

**Figure 1 biosensors-12-00690-f001:**
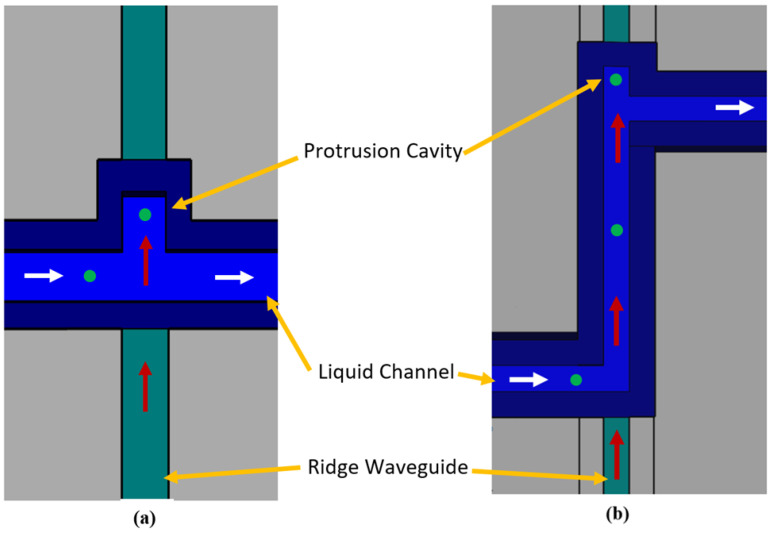
Optofluidic particle trapping for the orthogonal force (**a**) and gradient force (**b**) designs. Radiation pressure is shown by red arrows, fluid flow by white arrows, and green circles represent particles. The ridge waveguide, as shown by the green channel, couples light from a laser source off chip into the liquid channel where radiation pressure is used to push particles into the protrusion cavity. The protrusion cavity offers a physical mechanism to isolate the particles from the liquid flow.

**Figure 2 biosensors-12-00690-f002:**
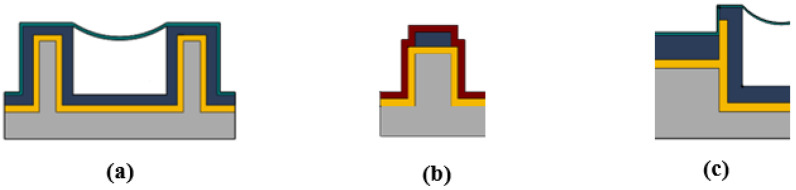
(**a**) Optofluidic channel with suspended nanomembrane. (**b**) Solid core ridge waveguide. (**c**) Interface between solid core ridge waveguide and liquid channel. The interface is thermal silicon dioxide. The ridge waveguide couples light into liquid channel for particle manipulation. Thermal silicon dioxide is shown in gold (*n* = 1.44), high refractive index PECVD silicon dioxide is shown in blue (*n* = 1.51), low refractive index PECVD silicon dioxide is shown in red (*n* = 1.44), PECVD silicon dioxide for the membrane is shown in light blue (*n* = 1.51), and silicon is shown in grey.

**Figure 3 biosensors-12-00690-f003:**
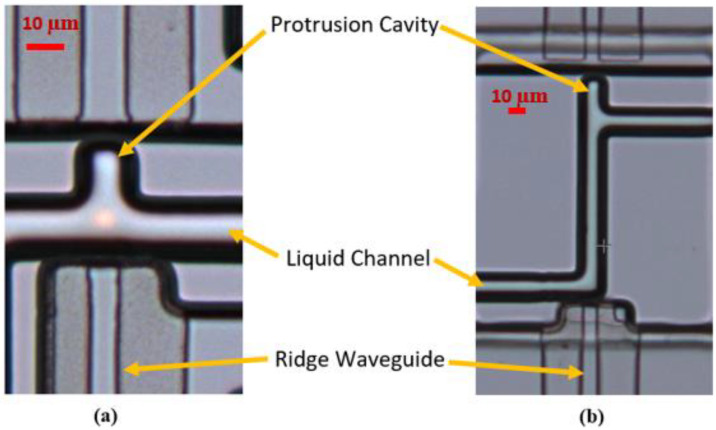
Optical manipulation using the orthogonal force design (**a**) and the gradient force design (**b**). The orthogonal force design introduces radiation pressure perpendicular to the fluid stream and has a short (~10 μm) optofluidic section. The gradient force design introduces light in parallel to the fluid flow and has an optofluidic focusing section (~100 μm) before the protrusion cavity. Images taken with an optical microscope.

**Figure 4 biosensors-12-00690-f004:**
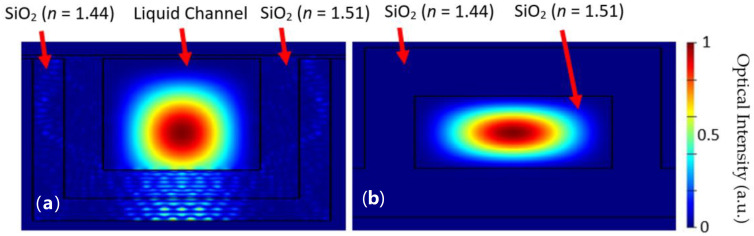
E field intensity zero order mode profile of the liquid channel (**a**) and ridge waveguide (**b**) using Lumerical FDE with a 532 nm wavelength.

**Figure 5 biosensors-12-00690-f005:**
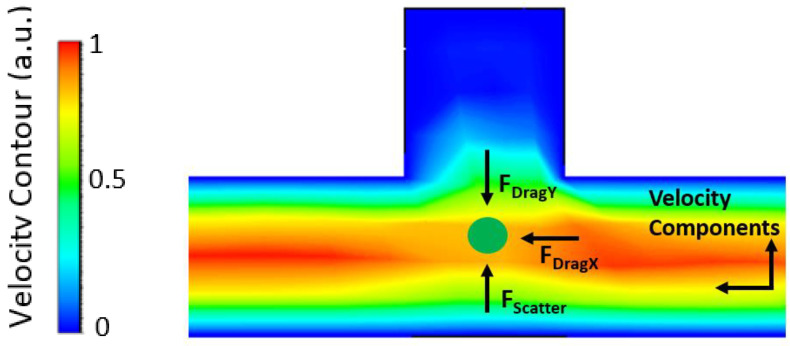
Forces acting on a microbead (green circle) in the orthogonal force design. Fluid simulation was performed in Ansys Fluent. The color spectrum corresponds to various flow velocities with red being high and blue being low.

**Figure 6 biosensors-12-00690-f006:**
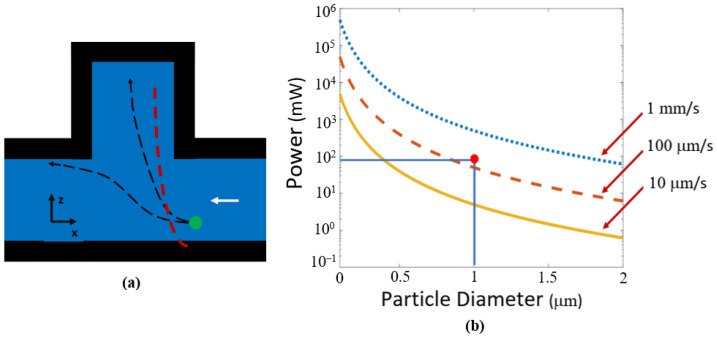
Orthogonal force design trapping characteristics. (**a**) Illustration of possible paths for a particle (green circle) passing a protrusion cavity showing a possible trapping event and an escape event demonstrated by the black dotted arrows. An approximate actual particle trajectory taken from Appendix A is shown by the red dotted line. The white arrow represents flow direction. (**b**) Optical trapping power needed to push particles into protrusion cavity, calculated for different particle sizes and fluid flow rates. The intersecting lines and dot on the plot demonstrate the power and particle diameter used experimentally.

**Figure 7 biosensors-12-00690-f007:**
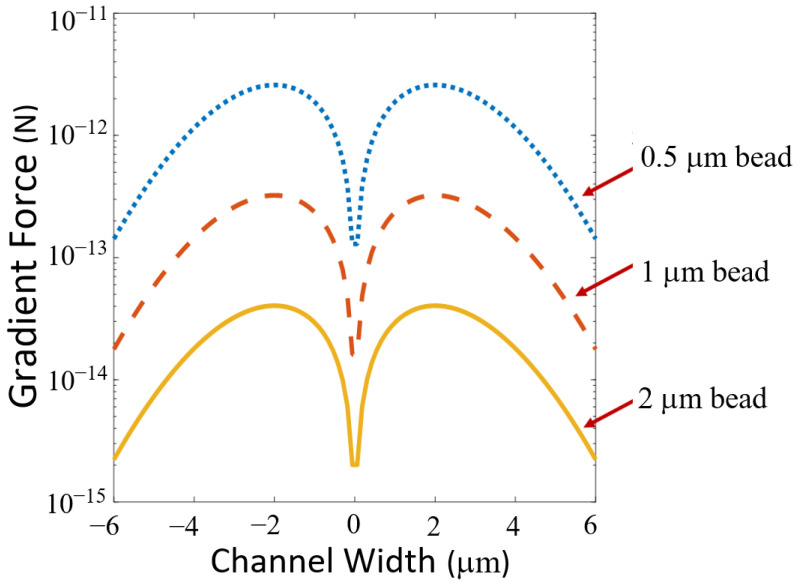
Gradient force versus channel position in the liquid channel for various particle diameters. The force intensity increases from the side wall as it approaches the center. The change in the force is relatively low in the center where the power is the highest and is evenly distributed.

**Figure 8 biosensors-12-00690-f008:**
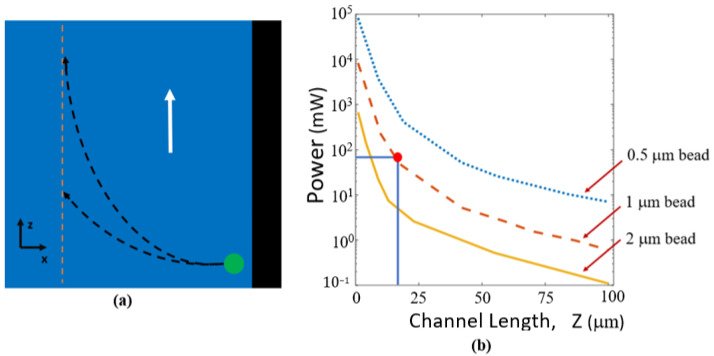
(**a**) Illustration of possible particle paths (black dotted arrows) for the worst-case scenario when a particle is at the edge of the channel at a side wall. The magnitude of the gradient force will determine how quickly a particle will become centralized. The particle is represented by the green circle. The channel center is shown by the orange dotted line. Fluid flow is in the direction of the white arrow. (**b**) Optical power required for minimum centralizing gradient force versus channel length for various particle diameters based on Equations (2) and (3). Solving Equation (3) provides solutions that can be used to determine how far along the channel in the z direction the particle will travel before becoming centralized. The intersecting lines and dot representing the conditions used when experimentally testing the platform with a 1 μm bead.

**Figure 9 biosensors-12-00690-f009:**
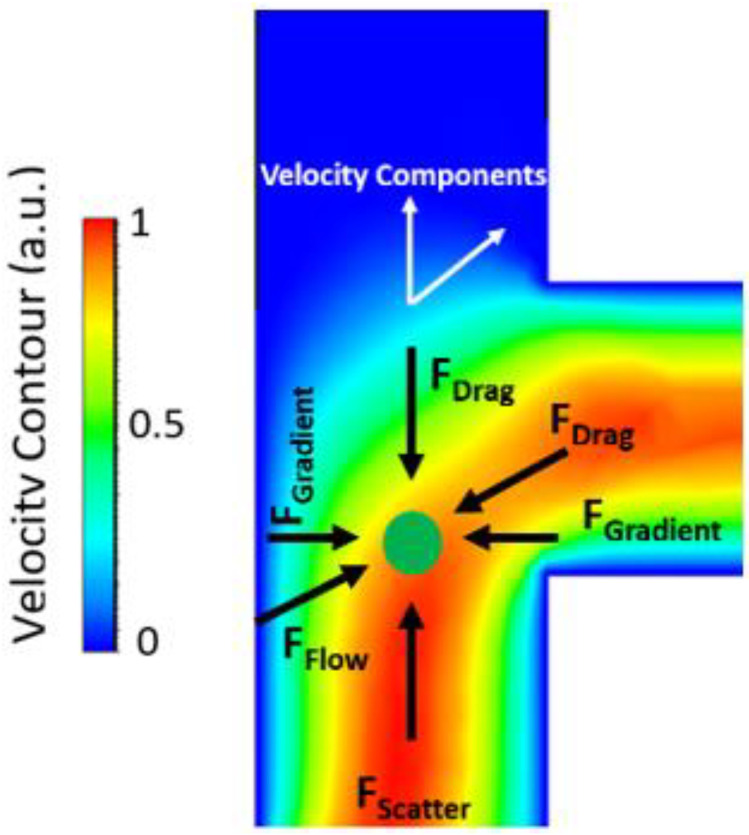
Illustration of forces acting on a particle in the optofluidic region at the channel bend simulated in Ansys Fluent. The scattering force direct the particle towards the protrusion cavity while the gradient force contains the particle in the center of the channel. The fluid velocity tries to pull the particle out of the optofluidic region. The color spectrum demonstrates various fluid velocities with red being fast and blue being slow.

**Figure 10 biosensors-12-00690-f010:**
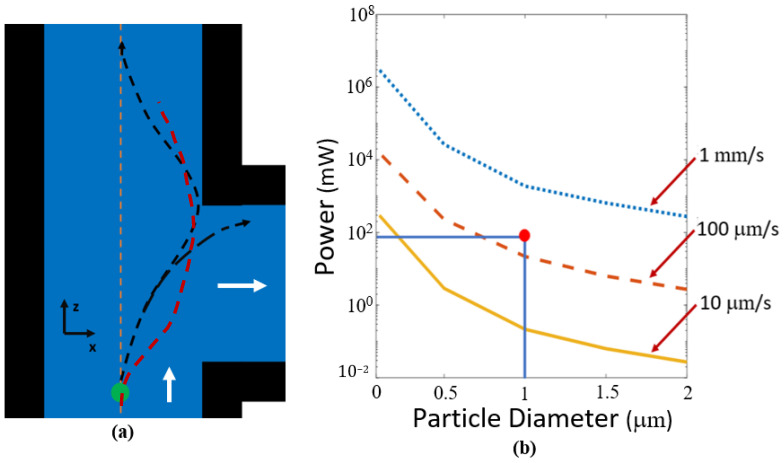
(**a**) Illustration of particles becoming susceptible to the fluid flow and either exiting the optofluidic region or becoming trapped. This is determined by the magnitude of the gradient force at the channel bend. These particle trajectories (black dotted arrows) closely resemble solutions to the differential equation (Equation (3)). An approximate actual particle trajectory taken from Appendix A shown by the red dotted line. The white arrows represent fluid flow direction and the orange dotted line marks the center of the channel. (**b**) Minimum trapping power versus particle size for various flow velocities at the optofluidic bend. The intersecting lines and dot demonstrate the experimental conditions for our tests.

**Figure 11 biosensors-12-00690-f011:**
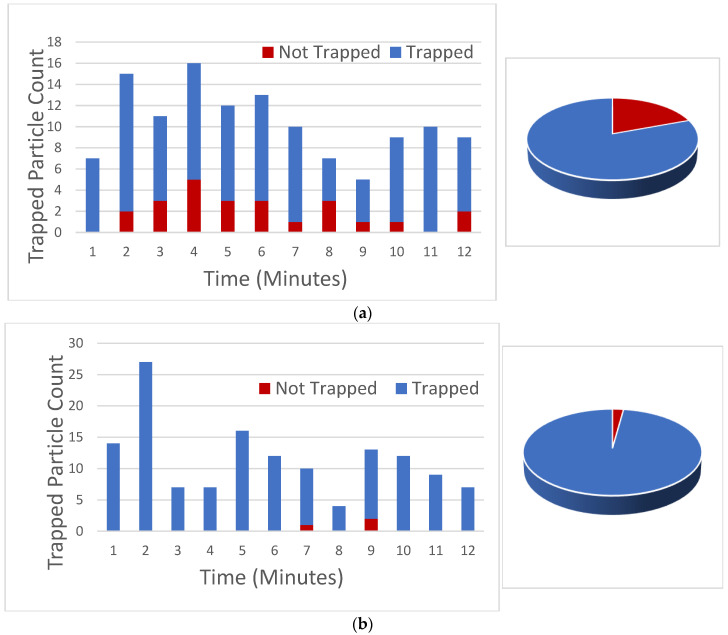
Trapping efficiency experiment for orthogonal force design (**a**) and gradient force design (**b**) over a given time interval. Particles trapped are recorded in blue while non-trapped particles are shown in red.

## Data Availability

Data is contained within the article.

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
