# Peer review of "Optofluidic Particle Manipulation: Optical Trapping in a Thin-Membrane Microchannel"

_biosensors, 2022, doi:10.3390/bios12090690_

Round 1

Reviewer 1 Report

In the manuscript "Optofluidic Particle Manipulation: [...]", the authors present an integrated microfluidic/optical trapping device designed to move particles of interest out of the fluid flowfield, with potential applications being particle sorting and localization.

While generally of high quality, the manuscript does suffer from several flaws that must be corrected prior to publication, primarily stemming from the simplified modeling but also including insufficient evidence.

1) The scattering and gradient force equations (1) and (2) hold only for homogeneous spheres. I understand that more advanced modeling to incorporate a variety of particle shapes and compositions is not within the scope of this manuscript, but at the very least the authors should clearly spell out their model's limitation(s).  For example, the particles' index of refraction was stated to be 1.57.  How was this determined?  What about the refractive index of the fluid?

2) The claim on line 198-199, "If the velocity induced on the particle by the scattering force..." seems weak since the fluid flow velocity varies within the flow channel, and in addition the scattering force acts only in 'y', while the fluid drag is mostly in 'x'.   Similarly, does the drag force calculation shown in Fig 6 hold for the confined flow chamber geometry?  Perhaps it does, but the authors need to provide some discussion.

3) The transmission efficiency (19%), mentioned on line 209 and again on line 311 (for 17.9%) seems based purely on a calculation and was not measured.  It seems to me that a measurement here is critical, since the fabricated device may have losses that are not included in the calculations.

4) It is unclear (to me) that the gradient force has any role in Video 2; the scattering force alone could account for the observed behavior.

5) In general, the videos are insufficient evidence; Video 1 in particular is very unclear, and as I mentioned Video 2 could simply result from an optical scattering force.  In addition, Video 2 has an audio track but is unclear what is being said or its relevance to the manuscript.

6) There were some formatting problems with the manuscript- many of the references are instead "Error! Reference source not found".  I understand this is a (somewhat) minor concern at this point in the process.

7) Figure 11 is unclear- why not simply provide the percentage of trapped microspheres, especially since the authors claim that the experimental data 'closely matches' predictions with 2 significant figures?  Why is there a time series?  Do the authors anticipate changes due to (perhaps) thermal effects?

In conclusion, while the authors have provided an interesting concept and some preliminary results, the evidence presented is insufficient to support or refute the claims made by the authors.  Perhaps some video taken using a laser power less than the minimum trapping power would be helpful, especially if the laser power could be adjusted during video acquisition.

Reviewer 2 Report

The manuscript of biosensors-1860604Optofluidic Particle Manipulation: Optical Trapping in a Thin-Membrane Microchannel” for Biosensors proposed a new approach for the utilization of particle trapping in microfluidic channels using optical trapping technique. The authors provide valuable clues to capture of target particles by changing the design. Additionally, the efficiency of trapping performance can reach up to 98%. Therefore, after carefully reading through the manuscript, this manuscript would be suitable to be publish in this journal after minor revision.

The comments are as following:

1.     The authors should revise the manuscript. It must be some kind of program failure and appear “Error! Reference source not found” all over the manuscript. Additionally, the figure caption did not show in the manuscript.

2.     The image quality of Figure 11 should be upgraded with higher quality of images.

Reviewer 3 Report

The authors demonstrate use of optical trapping in flow using two different designs. Fluid flow and electrical field simulations are carried out to better understand the power/efficiency requirements. This can be an interesting topic but there are major issues with this paper as described below. 

1) The organization of the paper is extremely poor. Please rewrite the paper in standard format. Include an experimental and methods section that describes materials, fabrication methods, so on and so forth. Computation simulations are still "results". 

2) Paper lacks experimental and scientific vigor. Large part of results are computation without comparison to actual experiments. Fig 10, for example, should also have an experimental trajectory of particles. Fig 5 and 9 don't even have computational values of drag/scatter and other forces, so what really is the purpose of those figures? 

3) Why are trapping efficiencies not calculated as a function of different power? There is not experimental control for Fig 11.

4) Fig 11 looks like just a result from single experiment. There is no statistics, chip to chip variability or details about how the data was collected. 

5) There are many "Error! Reference source not found" throughout the text. 

Round 2

Reviewer 1 Report

All my previous comments were properly addressed with the exception of Item 6 (formatting issues).  I will leave this to the professionals.  I have no science concerns remaining- nice job!

Reviewer 3 Report

The changes to my comments are satisfactory.